# Diversity Can Be Transferred: Output Diversification for White- and Black-box Attacks

**Yusuke Tashiro[123*], Yang Song[1], Stefano Ermon[1]**
[1]Department of Computer Science, Stanford University, Stanford, CA, USA
[2]Mitsubishi UFJ Trust Investment Technology Institute, Tokyo, Japan
[3]Japan Digital Design, Tokyo, Japan
{ytashiro,songyang,ermon}@stanford.edu

## Abstract

Adversarial attacks often involve random perturbations of the inputs drawn from uniform or Gaussian distributions, e.g., to initialize optimization-based white-box attacks or generate update directions in black-box attacks. These simple perturbations, however, could be sub-optimal as they are agnostic to the model being attacked. To improve the efficiency of these attacks, we propose Output Diversified Sampling (ODS), a novel sampling strategy that attempts to maximize diversity in the target model's outputs among the generated samples. While ODS is a gradient-based strategy, the diversity offered by ODS is transferable and can be helpful for both white-box and black-box attacks via surrogate models. Empirically, we demonstrate that ODS significantly improves the performance of existing white-box and black-box attacks. In particular, ODS reduces the number of queries needed for state-of-the-art black-box attacks on ImageNet by a factor of two.

## 1 Introduction

Deep neural networks have achieved great success in image classification. However, it is known that they are vulnerable to adversarial examples [1] — small perturbations imperceptible to humans that cause classifiers to output wrong predictions. Several studies have focused on improving model robustness against these malicious perturbations. Examples include adversarial training [2, 3], input purification using generative models [4, 5], regularization of the training loss [6, 7, 8, 9], and certified defenses [10, 11, 12].

Strong attacking methods are crucial for evaluating the robustness of classifiers and defense mechanisms. Many existing adversarial attacks rely on random sampling, i.e., adding small random noise to the input. In white-box settings, random sampling is widely used for random restarts [13, 14, 15, 16] to find a diverse set of starting points for the attacks. Some black-box attack methods also use random sampling to explore update directions [17, 18] or to estimate gradients of the target models [19, 20, 21]. In these attacks, random perturbations are typically sampled from a naïve uniform or Gaussian distribution in the input pixel space.

Random sampling in the input space, however, may not sufficiently explore the output (logits) space of a neural network — diversity in the input space does not directly translate to diversity in the output space of a deep nonlinear model. We illustrate this phenomenon in the left panel of Figure 1. When we add random perturbations to an image in the input space (see dashed blue arrows in the first plot of Figure 1), the corresponding output logits could be very similar to the output for the original image (as illustrated by the second plot of Figure 1). Empirically, we observe that this phenomenon can negatively impact the performance of attack methods.

To overcome this issue, we propose a sampling strategy designed to obtain samples that are diverse in the output space. Our idea is to perturb an input away from the original one as measured directly by

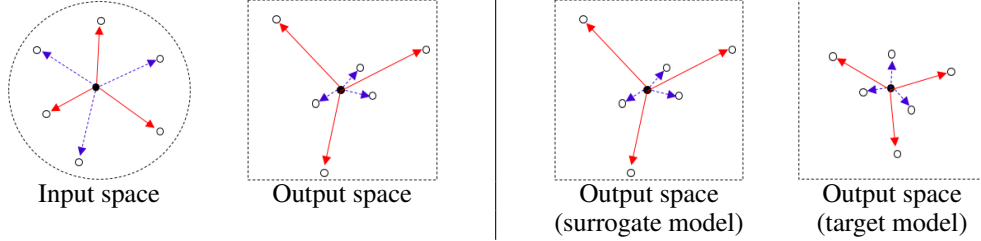

| Input space | Output space | Output space (surrogate model) | Output space (target model) |

Figure 1: Illustration of the differences between random sampling (blue dashed arrows) and ODS (red solid arrows). In each figure, the black 'o' corresponds to an original image, and white 'o's represent sampled perturbations. (Left): white-box setting. Perturbations by ODS in the input space are crafted by maximizing the distance in the output space. (Right): black-box setting. Perturbations crafted on the surrogate model transfer well to perturbations on the target model.

distances in the output space (see solid red arrows in the second plot in Figure 1). First, we randomly specify a direction in the output space. Next, we perform gradient-based optimization to generate a perturbation in the input space that yields a large change in the specified direction. We call this new sampling technique Output Diversified Sampling (ODS).

ODS can improve adversarial attacks under both white-box and black-box settings. For white-box attacks, we exploit ODS to initialize the optimization procedure of finding adversarial examples (called ODI). ODI typically provides much more diverse (and effective) starting points for adversarial attacks. Moreover, this initialization strategy is agnostic to the underlying attack method, and can be incorporated into most optimization-based white-box attack methods. Empirically, we demonstrate that ODI improves the performance of $\ell_\infty$ and $\ell_2$ attacks compared to naïve initialization methods. In particular, the PGD attack with ODI outperforms the state-of-the-art MultiTargeted attack [16] against pre-trained defense models, with 50 times smaller computational complexity on CIFAR-10.

In black-box settings, we cannot directly apply ODS because we do not have access to gradients of the target model. As an alternative, we apply ODS to surrogate models and observe that the resulting samples are diverse with respect to the target model: diversity in the output space transfers (see the rightmost two plots in Figure 1). Empirically, we demonstrate that ODS can reduce the number of queries needed for a score-based attack (SimBA [18]) by a factor of two on ImageNet. ODS also shows better query-efficiency than P-RGF [22], which is another method exploiting surrogate models to improve a black-box attack. These attacks with ODS achieve better query-efficiency than the state-of-the-art Square Attack [23]. In addition, ODS with a decision-based attack (Boundary Attack [17]) reduces the median perturbation distances of adversarial examples by a factor of three compared to the state-of-the-art HopSkipJump [24] and Sign-OPT [25] attacks.

## 2 Preliminaries

We denote an image classifier as $\mathbf{f} : \mathbf{x} \in [0,1]^D \mapsto \mathbf{z} \in \mathbb{R}^C$, where $\mathbf{x}$ is an input image, $\mathbf{z}$ represents the logits, and $C$ is the number of classes. We use $h(\mathbf{x}) = \arg\max_{c=1,\ldots,C} f_c(\mathbf{x})$ to denote the model prediction, where $f_c(\mathbf{x})$ is the $c$-th element of $\mathbf{f}(\mathbf{x})$.

Adversarial attacks can be classified into targeted and untargeted attacks. Given an image $\mathbf{x}$, a label $y$ and a classifier $\mathbf{f}$, the purpose of untargeted attacks is to find an adversarial example $\mathbf{x}^{\mathrm{adv}}$ that is similar to $\mathbf{x}$ but causes misclassification $h(\mathbf{x}^{\mathrm{adv}}) \neq y$. In targeted settings, attackers aim to change the model prediction $h(\mathbf{x}^{\mathrm{adv}})$ to a particular target label $t \neq y$. The typical goal of adversarial attacks is to find an adversarial example $\mathbf{x}^{\mathrm{adv}}$ within $B_\epsilon(\mathbf{x}) = \{\mathbf{x} + \boldsymbol{\delta} : \|\boldsymbol{\delta}\|_p \leq \epsilon\}$, i.e., the $\epsilon$-radius ball around an original image $\mathbf{x}$. Another common setting is to find a valid adversarial example with the smallest $\ell_p$ distance from the original image.

**White-box attacks** In white-box settings, attackers can access full information of the target model. One strong and popular example is the Projected Gradient Descent (PGD) attack [2], which iteratively applies the following update rule:

$$\mathbf{x}_{k+1}^{\mathrm{adv}} = \mathrm{Proj}_{B_\epsilon(\mathbf{x})} \left( \mathbf{x}_k^{\mathrm{adv}} + \eta \, \mathrm{sign} \left( \nabla_{\mathbf{x}_k^{\mathrm{adv}}} L(\mathbf{f}(\mathbf{x}_k^{\mathrm{adv}}), y) \right) \right) \tag{1}$$

where $\operatorname{Proj}_{B_\epsilon(\mathbf{x})}(\mathbf{x}^{\text{adv}}) \triangleq \arg\min_{\mathbf{x}' \in B_\epsilon(\mathbf{x})} \|\mathbf{x}^{\text{adv}} - \mathbf{x}'\|_p$, $\eta$ is the step size, and $L(\mathbf{f}(\mathbf{x}), y)$ is a loss function, e.g. the margin loss defined as $\max_{i \neq y} f_i(\mathbf{x}) - f_y(\mathbf{x})$. To increase the odds of success, the procedure is restarted multiple times with uniformly sampled initial inputs from $B_\epsilon(\mathbf{x})$.

**Black-box attacks**  In black-box settings, the attacker only has access to outputs of the target model without knowing its architecture and weights. Black-box attacks can be largely classified into transfer-based, score-based, and decision-based methods respectively. Transfer-based attacks craft white-box adversarial examples with respect to surrogate models, and transfer them to the target model. The surrogate models are typically trained with the same dataset as the target model so that they are close to each other. In score-based settings, attackers can know the output scores (logits) of the classifier; while for decision-based settings, attackers can only access the output labels of the classifier. For these two approaches, attacks are typically evaluated in terms of query efficiency, i.e. the number of queries needed to generate an adversarial example and its perturbation size.

Recently, several studies [22, 26, 27] employed surrogate models to estimate the gradients of the loss function of the target model. Some attack methods used random sampling in the input space, such as the decision-based Boundary Attack [17] and the score-based Simple Black-Box Attack [18].

## 3 Output Diversified Sampling

As intuitively presented in Figure 1, random sampling in the input space does not necessarily produce samples with high diversity as measured in the output space. To address this problem, we propose Output Diversified Sampling (ODS). Given an image $\mathbf{x}$, a classifier $\mathbf{f}$ and the direction of diversification $\mathbf{w}_d \in \mathbb{R}^C$, we define the normalized perturbation vector of ODS as follows:

$$\mathbf{v}_{\text{ODS}}(\mathbf{x}, \mathbf{f}, \mathbf{w}_d) = \frac{\nabla_{\mathbf{x}}(\mathbf{w}_d^\mathsf{T}\mathbf{f}(\mathbf{x}))}{\|\nabla_{\mathbf{x}}(\mathbf{w}_d^\mathsf{T}\mathbf{f}(\mathbf{x}))\|_2}, \tag{2}$$

where $\mathbf{w}_d$ is sampled from the uniform distribution over $[-1, 1]^C$. Below we show how to enhance white- and black-box attacks with ODS.

### 3.1 Initialization with ODS for white-box attacks

In white-box settings, we utilize ODS for initialization (ODI) to generate output-diversified starting points. Given an original input $\mathbf{x}_{\text{org}}$ and the direction for ODS $\mathbf{w}_d$, we try to find a restart point $\mathbf{x}$ that is as far away from $\mathbf{x}_{\text{org}}$ as possible by maximizing $\mathbf{w}_d^\mathsf{T}(\mathbf{f}(\mathbf{x}) - \mathbf{f}(\mathbf{x}_{\text{org}}))$ via the following iterative update:

$$\mathbf{x}_{k+1} = \operatorname{Proj}_{B(\mathbf{x}_{\text{org}})} (\mathbf{x}_k + \eta_{\text{ODI}} \operatorname{sign}(\mathbf{v}_{\text{ODS}}(\mathbf{x}_k, \mathbf{f}, \mathbf{w}_d))) \tag{3}$$

where $B(\mathbf{x}_{\text{org}})$ is the set of allowed perturbations, which is typically an $\epsilon$-ball in $\ell_p$ norm, and $\eta_{\text{ODI}}$ is a step size. When applying ODI to $\ell_2$ attacks, we omit the sign function. After some steps of ODI, we start an attack from the restart point obtained by ODI. We sample a new direction $\mathbf{w}_d$ for each restart in order to obtain diversified starting points for the attacks. We provide the pseudo-code for ODI in Algorithm A of the Appendix.

One sampling step of ODI costs roughly the same time as one iteration of most gradient-based attacks (e.g., PGD). Empirically, we observe that the number of ODI steps $N_{\text{ODI}} = 2$ is already sufficient to obtain diversified starting points (details of the sensitivity analysis are in Appendix C.2), and fix $N_{\text{ODI}} = 2$ in all our experiments unless otherwise specified. We emphasize that ODS is not limited to PGD, and can be applied to a wide family of optimization-based adversarial attacks.

**Experimental verification of increased diversity:** We quantitatively evaluate the diversity of starting points in terms of pairwise distances of output values $\mathbf{f}(\mathbf{x})$, confirming the intuition presented in the left two plots of Figure 1. We take a robust model on CIFAR-10 as an example of target models, and generate starting points with both ODI and standard uniform initialization to calculate the mean pairwise distance. The pairwise distance (i.e. diversity) obtained by ODI is 6.41, which is about 15 times larger than that from uniform initialization (0.38). In addition, PGD with the same steps as ODI does not generate diverse samples (pairwise distance is 0.43). Details are in Appendix C.1.

---

**Algorithm 1** Simple Black-box Attack [18] with ODS

---

1: **Input:** A targeted image $\mathbf{x}$, loss function $L$, a target classifier $\mathbf{f}$, a set of surrogate models $\mathcal{G}$
2: **Output:** attack result $\mathbf{x}_{\text{adv}}$
3: Set the starting point $\mathbf{x}_{\text{adv}} = \mathbf{x}$
4: **while** $\mathbf{x}_{\text{adv}}$ is not adversary **do**
5:     Choose a surrogate model $\mathbf{g}$ from $\mathcal{G}$, and sample $\mathbf{w}_{\text{d}} \sim U(-1,1)^C$
6:     Set $\mathbf{q} = \mathbf{v}_{\text{ODS}}(\mathbf{x}_{\text{adv}}, \mathbf{g}, \mathbf{w}_{\text{d}})$
7:     **for** $\alpha \in \{\epsilon, -\epsilon\}$ **do**
8:         **if** $L(\mathbf{x}_{\text{adv}} + \alpha \cdot \mathbf{q}) > L(\mathbf{x}_{\text{adv}})$ **then**
9:             Set $\mathbf{x}_{\text{adv}} = \mathbf{x}_{\text{adv}} + \alpha \cdot \mathbf{q}$ and **break**

---

## 3.2 Sampling update directions with ODS for black-box attacks

In black-box settings, we employ ODS to sample update directions instead of random sampling. Given a target classifier $\mathbf{f}$, we cannot directly calculate the ODS perturbation $\mathbf{v}_{\text{ODS}}(\mathbf{x}, \mathbf{f}, \mathbf{w}_{\text{d}})$ because gradients of the target model $\mathbf{f}$ are unknown. Instead, we introduce a surrogate model $\mathbf{g}$ and calculate the ODS vector $\mathbf{v}_{\text{ODS}}(\mathbf{x}, \mathbf{g}, \mathbf{w}_{\text{d}})$.

ODS can be applied to attack methods that rely on random sampling in the input space. Since many black-box attacks use random sampling to explore update directions [17, 18] or estimate gradients of the target models [19, 21, 24], ODS has broad applications. In this paper, we apply ODS to two popular black-box attacks that use random sampling: decision-based Boundary Attack [17] and score-based Simple Black-Box Attack (SimBA [18]). In addition, we compare ODS with P-RGF [22], which is an another attack method using surrogate models.

To illustrate how we apply ODS to existing black-box attack methods, we provide the pseudo-code of SimBA [18] with ODS in Algorithm 1. The original SimBA algorithm picks an update direction $\mathbf{q}$ randomly from a group of candidates $Q$ that are orthonormal to each other. We replace it with ODS, as shown in the line 5 and 6 of Algorithm 1. For other attacks, we replace random sampling with ODS in a similar way. Note that in Algorithm 1, we make use of multiple surrogate models and uniformly sample one each time, since we empirically found that using multiple surrogate models can make attacks stronger.

**Experimental verification of increased diversity:** We quantitatively evaluate that ODS can lead to high diversity in the output space of the target model, as shown in the right two plots of Figure 1. We use pre-trained Resnet50 [28] and VGG19 [29] models on ImageNet as the target and surrogate models respectively. We calculate and compare the mean pairwise distances of samples with ODS and random Gaussian sampling. The pairwise distance (i.e. diversity) for ODS is 0.79, which is 10 times larger than Gaussian sampling (0.07). Details are in Appendix D.1. We additionally observe that ODS does not produce diversified samples when we use random networks as surrogate models. This indicates that good surrogate models are crucial for transferring diversity.

# 4 Experiments in white-box settings

In this section, we show that the diversity offered by ODI can improve white-box attacks for both $\ell_\infty$ and $\ell_2$ distances. Moreover, we demonstrate that a simple combination of PGD and ODI achieves new state-of-the-art attack success rates. All experiments are for untargeted attacks.

## 4.1 Efficacy of ODI for white-box attacks

We combine ODI with two popular attacks: PGD attack [2] with the $\ell_\infty$ norm and C&W attack [30] with the $\ell_2$ norm. We run these attacks on MNIST, CIFAR-10 and ImageNet.

**Setup** We perform attacks against three adversarially trained models from MadryLab[1] [2] for MNIST and CIFAR-10 and the Feature Denoising ResNet152 network[2] [31] for ImageNet. For PGD

attacks, we evaluate the model accuracy with 20 restarts, where starting points are uniformly sampled over an $\epsilon$-ball for the naïve resampling. For C&W attacks, we calculate the minimum $\ell_2$ perturbation that yields a valid adversarial example among 10 restarts for each image, and measure the average of the minimum perturbations. Note that the original paper of C&W [30] attacks did not apply random restarts. Here for the naïve initialization of C&W attacks we sample starting points from a Gaussian distribution and clip it into an $\epsilon$-ball (details in Appendix B.1).

For fair comparison, we test different attack methods with the same amount of computation. Specifically, we compare $k$-step PGD with naïve initialization (denoted as PGD-$k$) against ($k$-2)-step PGD with 2-step ODI (denoted as ODI-PGD-($k$-2)). We do not adjust the number of steps for C&W attacks because the computation time of 2-step ODI are negligible for C&W attacks.

Table 1: Comparing different white-box attacks. We report model accuracy (lower is better) for PGD and average of the minimum $\ell_2$ perturbations (lower is better) for C&W. All results are the average of three trials.

| model | PGD | | C&W | |
|---|---|---|---|---|
| | naïve (PGD-$k$) | ODI (ODI-PGD-($k$-2)) | naïve | ODI |
| MNIST | $90.31 \pm 0.02\%$ | $\mathbf{90.21} \pm 0.05\%$ | $2.27 \pm 0.00$ | $\mathbf{2.25} \pm 0.01$ |
| CIFAR-10 | $46.06 \pm 0.02\%$ | $\mathbf{44.45} \pm 0.02\%$ | $0.71 \pm 0.00$ | $\mathbf{0.67} \pm 0.00$ |
| ImageNet | $43.5 \pm 0.0\%$ | $\mathbf{42.3} \pm 0.0\%$ | $1.58 \pm 0.00$ | $\mathbf{1.32} \pm 0.01$ |

**Results**  We summarize all quantitative results in Table 1. Attack performances with ODI are better than naïve initialization for all models and attacks. The improvement by ODI on the CIFAR-10 and ImageNet models is more significant than on the MNIST model. We hypothesize that this is due to the difference in model non-linearity. When a target model includes more non-linear transformations, the difference in diversity between the input and output space could be larger, in which case ODI will be more effective in providing a diverse set of restarts.

### 4.2 Comparison between PGD attack with ODI and state-of-the-art attacks

To further demonstrate the power of ODI, we perform ODI-PGD against MadryLab's robust models [2] on MNIST and CIFAR-10 and compare ODI-PGD with state-of-the-art attacks.

**Setup**  One state-of-the-art attack we compare with is the well-tuned PGD attack [16], which achieved 88.21% accuracy for the robust MNIST model. The other attack we focus on is the MultiTargeted attack [16], which obtained 44.03% accuracy against the robust CIFAR-10 model. We use all test images on each dataset and perform ODI-PGD under two different settings. One is the same as Section 4.1. The other is ODI-PGD with tuned hyperparameters, e.g. increasing the number of steps and restarts. Please see Appendix B.2 for more details of tuning.

Table 2: Comparison of ODI-PGD with state-of-the-art attacks against pre-trained defense models. The complexity rows display products of the number of steps and restarts. Results for ODI-PGD are the average of three trials. For ODI-PGD, the number of steps is the sum of ODS and PGD steps.

| model | | ODI-PGD (in Sec. 4.1) | tuned ODI-PGD | tuned PGD [16] | MultiTargeted [16] |
|---|---|---|---|---|---|
| MNIST | accuracy | $90.21 \pm 0.05\%$ | $\mathbf{88.13} \pm 0.01\%$ | 88.21% | 88.36% |
| | complexity | $40 \times 20$ | $1000 \times 1000$ | $1000 \times 1800$ | $1000 \times 1800$ |
| CIFAR-10 | accuracy | $44.45 \pm 0.02\%$ | $\mathbf{44.00} \pm 0.01\%$ | 44.51% | 44.03% |
| | complexity | $20 \times 20$ | $150 \times 20$ | $1000 \times 180$ | $1000 \times 180$ |

**Results**  We summarize the comparison between ODI-PGD and state-of-the-art attacks in Table 2. Our tuned ODI-PGD reduces the accuracy to 88.13% for the MNIST model, and to 44.00% for the CIFAR-10 model. These results outperform existing state-of-the-art attacks.

To compare their running time, we report the total number of steps (the number of steps multiplied by the number of restarts) as a metric of complexity, because the total number of steps is equal to the number of gradient computations (the computation time per gradient evaluation is comparable for all gradient-based attacks). In Table 2, the computational cost of tuned ODI-PGD is smaller than that of state-of-the-art attacks, and especially 50 times smaller on CIFAR-10. Surprisingly, even without tuning ODI-PGD (in the first column) can still outperform tuned PGD [16] while also being drastically more efficient computationally.

## 5 Experiments in black-box settings

In this section, we demonstrate that black-box attacks combined with ODS significantly reduce the number of queries needed to generate adversarial examples. In experiments below, we randomly sample 300 correctly classified images from the ImageNet validation set. We evaluate both untargeted and targeted attacks. For targeted attacks, we uniformly sample target labels.

### 5.1 Query-efficiency of score-based attacks with ODS

#### 5.1.1 Applying ODS to score-based attacks

To show the efficiency of ODS, we combine ODS with the score-based Simple Black-Box Attack (SimBA) [18]. SimBA randomly samples a vector and either adds or subtracts the vector to the target image to explore update directions. The vector is sampled from a pre-defined set of orthonormal vectors in the input space. These are the discrete cosine transform (DCT) basis vectors in the original paper [18]. We replace the DCT basis vectors with ODS sampling (called SimBA-ODS),

**Setup**  We use pre-trained ResNet50 model as the target model and select four pre-trained models (VGG19, ResNet34, DenseNet121 [32], MobileNetV2 [33]) as surrogate models. We set the same hyperparameters for SimBA as [18]: the step size is 0.2 and the number of iterations (max queries) is 10000 (20000) for untargeted attacks and 30000 (60000) for targeted attacks. As the loss function in SimBA, we employ the margin loss for untargeted attacks and the cross-entropy loss for targeted attacks.

**Results**  First, we compare SimBA-DCT [18] and SimBA-ODS. Table 3 reports the number of queries and the median $\ell_2$ perturbations. Remarkably, SimBA-ODS reduces the average number of queries by a factor between 2 and 3 compared to SimBA-DCT for both untargeted and targeted settings. This confirms that ODS not only helps white-box attacks, but also leads to significant improvements of query-efficiency in black-box settings. In addition, SimBA-ODS decreases the average perturbation sizes by around a factor of two, which means that ODS helps find better adversarial examples that are closer to the original image.

Table 3: Number of queries and size of $\ell_2$ perturbations for score-based attacks.

| attack | num. of surrogates | untargeted | | | targeted | | |
|---|---|---|---|---|---|---|---|
| | | success rate | average queries | median $\ell_2$ perturbation | success rate | average queries | median $\ell_2$ perturbation |
| SimBA-DCT [18] | 0 | **100.0%** | 909 | 2.95 | 97.0% | 7114 | 7.00 |
| SimBA-ODS | 4 | **100.0%** | **242** | **1.40** | **98.3%** | **3503** | **3.55** |

#### 5.1.2 Comparing ODS with other methods using surrogate models

We consider another black-box attack that relies on surrogate models: P-RGF [22], which improves over the original RGF (random gradient-free) method for gradient estimation. P-RGF exploits prior knowledge from surrogate models to estimate the gradient more efficiently than RGF. Since RGF uses random sampling to estimate the gradient, we propose to apply ODS to RGF (new attack named ODS-RGF) and compare it with P-RGF under $\ell_2$ and $\ell_\infty$ norms.

For fair comparison, we use a single surrogate model as in [22]. We choose pre-trained ResNet50 model as the target model and ResNet34 model as the surrogate model. We give query-efficiency

Table 4: Comparison of ODS-RGF and P-RGF on ImageNet. Hyperparameters for RGF are same as [22] :max queries are 10000, sample size is 10, step size is 0.5 ($\ell_2$) and 0.005 ($\ell_\infty$), and epsilon is $\sqrt{0.001 \cdot 224^2 \cdot 3}$ ($\ell_2$) and 0.05 ($\ell_\infty$).

| | | | untargeted | | | targeted | | |
|---|---|---|---|---|---|---|---|---|
| norm | attack | num. of surrogates | success rate | average queries | median $\ell_2$ perturbation | success rate | average queries | median $\ell_2$ perturbation |
| | RGF | 0 | **100.0%** | 633 | 3.07 | **99.3%** | 3141 | 8.23 |
| $\ell_2$ | P-RGF [25] | 1 | **100.0%** | 211 | 2.08 | 97.0% | 2296 | 7.03 |
| | ODS-RGF | 1 | **100.0%** | **133** | **1.50** | **99.3%** | **1043** | **4.47** |
| | RGF | 0 | 97.0% | 520 | - | 25.0% | 2971 | - |
| $\ell_\infty$ | P-RGF [25] | 1 | 99.7% | 88 | - | 65.3% | 2123 | - |
| | ODS-RGF | 1 | **100.0%** | **74** | - | **92.0%** | **985** | - |

results of both methods in Table 4. The average number of queries required by ODS-RGF is less than that of P-RGF in all settings. This suggests ODS-RGF can estimate the gradient more precisely than P-RGF by exploiting diversity obtained via ODS and surrogate models. The differences between ODS-RGF and P-RGF are significant in targeted settings, since ODS-RGF achieves smaller perturbations than P-RGF (see median perturbation column). To verify the robustness of our results, we also ran experiments using VGG19 as a surrogate model and obtained similar results.

We additionally consider TREMBA [34], a black-box attack (restricted to the $\ell_\infty$-norm) that is state-of-the-art among those using surrogate models. In TREMBA, a low-dimensional embedding is learned via surrogate models so as to obtain initial adversarial examples which are then updated using a score-based attack. Our results show that ODS-RGF combined with SI-NI-DIM [35], which is a state-of-the-art transfer-based attack, is comparable to TREMBA even though ODS-RGF is not restricted to the $\ell_\infty$-norm. Results and more details are provided in Appendix D.3.

### 5.1.3 Comparison of ODS with state-of-the-art score-based attacks

To show the advantage of ODS and surrogate models, we compare SimBA-ODS and ODS-RGF with the Square Attack [23], which is a state-of-the-art attack for both $\ell_\infty$ and $\ell_2$ norms when surrogate models are not allowed. For comparison, we regard SimBA as $\ell_2$ bounded attacks: the attack is successful when adversarial $\ell_2$ perturbation is less than a given bound $\epsilon$. We set $\epsilon = 5$ ($\ell_2$) and 0.05 ($\ell_\infty$) as well as other hyperparameters according to the original paper [23], except that we set the max number of queries to 20000 for untargeted attacks and 60000 for targeted attacks. For ODS-RGF, we use four surrogate models as discussed in Section 5.1.1 for SimBA-ODS.

Table 5: Number of queries for attacks with ODS versus the Square Attack.

| | | | untargeted | | targeted | |
|---|---|---|---|---|---|---|
| norm | attack | num. of surrogates | success rate | average queries | success rate | average queries |
| | Square [23] | 0 | 99.7% | 647 | 96.7% | 11647 |
| $\ell_2$ | SimBA-ODS | 4 | 99.7% | 237 | 90.3% | 2843 |
| | ODS-RGF | 4 | **100.0%** | **144** | **99.0%** | **1285** |
| $\ell_\infty$ | Square [23] | 0 | **100.0 %** | **60** | **100.0%** | 2317 |
| | ODS-RGF | 4 | **100.0 %** | 78 | 97.7% | **1242** |

As shown in Table 5, the number of queries required for ODS-RGF and SimBA-ODS are lower than that of the Square Attack under the $\ell_2$ norm. The improvement is especially large for ODS-RGF. The difference between ODS-RGF and SimBA-ODS mainly comes from different base attacks (i.e., RGF and SimBA). For the $\ell_\infty$ norm setting, ODS-RGF is comparable to the Square Attack. We hypothesize that the benefit of estimated gradients by RGF decreases under the $\ell_\infty$ norm due to the sign function. However, because ODS can be freely combined with many base attacks, a stronger base attack is likely to further improve query-efficiency.

## 5.2 Query-efficiency of decision-based attacks with ODS

We demonstrate that ODS also improves query-efficiency for decision-based attacks. We combine ODS with the decision-based Boundary Attack [17]. The Boundary Attack starts from an image which is adversarial, and iteratively updates the image to find smaller perturbations. To generate the update direction, the authors of [17] sampled a random noise vector from a Gaussian distribution $\mathcal{N}(\mathbf{0}, \mathbf{I})$ each step. We replace this random sampling procedure with sampling by ODS (we call the new method Boundary-ODS). We give the pseudo-code of Boundary-ODS in Algorithm B (in the Appendix).

**Setup** We use the same settings as the previous section for score-based attacks: 300 validation images on ImageNet, pre-trained ResNet50 target model, and four pre-trained surrogate models. We test on both untargeted and targeted attacks. In targeted settings, we give randomly sampled images with target labels as initial images. We use the implementation in Foolbox [36] for Boundary Attack with default parameters, which is more efficient than the original implementation.

We also compare Boundary-ODS with two state-of-the-art decision-based attacks: the HopSkipJump attack [24] and the Sign-OPT attack [25]. We use the implementation in ART [37] for HopSkipJump and the author's implementation for Sign-OPT. We set default hyperparameters for both attacks.

**Results** Table 6 summarizes the median sizes of $\ell_2$ adversarial perturbations obtained with a fixed number of queries. Clearly, Boundary-ODS significantly improves query-efficiency compared to the original Boundary Attack. In fact, Boundary-ODS outperforms state-of-the-art attacks: it decreases the median $\ell_2$ perturbation at 10000 queries to less than one-third of previous best untargeted attacks and less than one-fourth of previous best targeted attacks. We additionally describe the relationship between median $\ell_2$ perturbations and the number of queries in Figure 2. Note that Boundary-ODS outperforms other attacks, especially in targeted settings. Moreover, Boundary-ODS only needs fewer than 3500 queries to achieve the adversarial perturbation obtained by other attacks with 10000 queries.

Table 6: Median $\ell_2$ perturbations for Boundary-ODS and decision-based state-of-the-art attacks.

| | | number of queries | | | | | |
| | | untargeted | | | targeted | | |
| attack | num. of surrogates | 1000 | 5000 | 10000 | 1000 | 5000 | 10000 |
|---|---|---|---|---|---|---|---|
| Boundary [17] | 0 | 45.07 | 11.46 | 4.30 | 73.94 | 41.88 | 27.05 |
| Boundary-ODS | 4 | **7.57** | **0.98** | **0.57** | **27.24** | **6.84** | **3.76** |
| HopSkipJump [24] | 0 | 14.86 | 3.50 | 1.79 | 65.88 | 33.98 | 18.25 |
| Sign-OPT [25] | 0 | 21.73 | 3.98 | 2.01 | 68.75 | 36.93 | 22.43 |

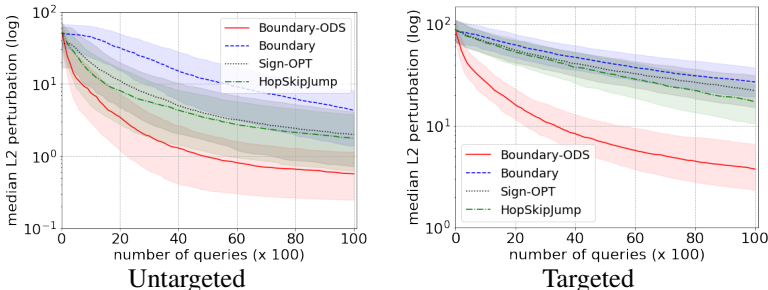

Figure 2: Relationship between median $\ell_2$ perturbations and the number of queries for decision-based attacks. Error bars show 25th and 75th percentile of $\ell_2$ perturbations.

## 5.3 Effectiveness of ODS with out-of-distribution images

Although several studies use prior knowledge from surrogate models to improve performance of black-box attacks, there is a drawback—those approaches require a dataset to train surrogate models.

In reality, it is typically impossible to obtain the same dataset used for training the target model. We show that ODS is applicable even when we only have a limited dataset that is out-of-distribution (OOD) and may contain only images with irrelevant labels.

We select 100 ImageNet classes which do not overlap with the classes used in the experiments of Section 5.2. We train surrogate models using an OOD training dataset with these 100 classes. We train five surrogate models with the same ResNet18 architecture because multiple surrogate models provide diversified directions. Then, we run Boundary-ODS with the trained surrogate models under the same setting as Section 5.2. As shown in Table 7, although Boundary-ODS with the OOD training dataset underperforms Boundary-ODS with the full dataset, it is still significantly better than the original Boundary Attack with random sampling. This demonstrates that the improved diversity achieved by ODS improves black-box attacks even if we only have OOD images to train a surrogate.

Table 7: Median $\ell_2$ perturbations for Boundary-ODS with surrogate models trained on OOD images.

| | number of queries | | | | | |
| | untargeted | | | targeted | | |
| attack | 1000 | 5000 | 10000 | 1000 | 5000 | 10000 |
|---|---|---|---|---|---|---|
| Boundary [17] | 45.07 | 11.46 | 4.30 | 73.94 | 41.88 | 27.05 |
| Boundary-ODS (OOD dataset) | **11.27** | **1.63** | **0.98** | **41.67** | **13.72** | **8.39** |
| Boundary-ODS (full dataset in Sec. 5.2) | 7.57 | 0.98 | 0.57 | 27.24 | 6.84 | 3.76 |

# 6 Related works

The closest approach to ours is the white-box MultiTargeted attack [16]. This attack changes the target class of attacks per restart, and it can be regarded as a method which aims to obtain more diversified starting points. However, MultiTargeted attack is limited to the setting of $\ell_p$-bounded white-box attacks. In contrast, ODS can be applied to more general white- and black-box attacks. In addition, ODS does not require the original class of the target image, therefore it is more broadly applicable. Further discussion are in Appendix E. Another related work is Interval Attack [38] which generates diverse starting points by leveraging symbolic interval propagation. While Interval Attack shows good performances against MNIST models, the attack is not scalable to large models.

ODS utilizes surrogate models, which are commonly used for black-box attacks. Most previous methods exploit surrogate models to estimate gradients of the loss function on the target model [39, 40, 41, 22, 26, 27]. Some recent works exploit surrogate models to train other models [42, 34] or update surrogate models during attacks [43]. Integrating ODS with these training-based methods is an interesting direction for future work.

# 7 Conclusion

We propose ODS, a new sampling strategy for white- and black-box attacks. By generating more diverse perturbations as measured in the output space, ODS can create more effective starting points for white-box attacks. Leveraging surrogate models, ODS also improves the exploration of the output space for black-box attacks. Moreover, ODS for black-box attacks is applicable even if the surrogate models are trained with out-of-distribution datasets. Therefore, black-box attacks with ODS are more practical than other black-box attacks using ordinary surrogate models. Our empirical results demonstrate that ODS with existing attack methods outperforms state-of-the-art attacks in various white-box and black-box settings.

While we only focus on ODS with surrogate models trained with labeled datasets, ODS may also work well using unlabeled datasets, which we leave as future work. One additional direction is to improve the efficiency of ODS by selecting suitable surrogate models with reinforcement learning.

## Broader Impact

The existence of adversarial examples is a major source of concern for machine learning applications in the real world. For example, imperceptible perturbations crafted by malicious attackers could deceive safety critical systems such as autonomous driving and facial recognition systems. Since adversarial examples exist not only for images, but also for other domains such as text and audio, the potential impact is large. Our research provides new state-of-the-art black-box adversarial attacks in terms of query-efficiency and makes adversarial attacks more practical and strong. While all experiments in this paper are for images, the proposed method is also applicable to other modalities. Because of this, our research could be used in harmful ways by malicious users.

On the positive side, strong attacks are necessary to develop robust machine learning models. For the last few years, several researchers have proposed adversarial attacks which break previous defense models. In response to these strong attacks, new and better defense mechanisms have been developed. It is this feedback loop between attacks and defenses that advances the field. Our research not only provides a state-of-the-art attack, but also sheds light on a new perspective, namely the importance of diversity, for improving adversarial attacks. This may have a long term impact on inspiring more effective defense methods.

## Acknowledgements and Disclosure of Funding

This research was supported in part by AFOSR (FA9550-19-1-0024), NSF (#1651565, #1522054, #1733686), ONR, and FLI.

## Footnotes

[1]`https://github.com/MadryLab/mnist_challenge` and `https://github.com/MadryLab/cifar10_challenge`. We use their secret model.

[2]`https://github.com/facebookresearch/ImageNet-Adversarial-Training`.

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
