[Supplementary Material]

# A Pseudo-code of proposed methods

In this section, we provide the pseudo-codes of methods proposed in the main paper. First, Algorithm A shows the pseudo-code of ODI for white-box attacks in Section 3.1. The line 4-6 in the algorithm describes the iterative update by ODI.

---
**Algorithm A** Initialization by ODS (ODI) for white-box attacks
---
1: **Input:** A targeted image $\mathbf{x}_{org}$, a target classifier $\mathbf{f}$, perturbation set $B(\mathbf{x}_{org})$, number of ODI steps $N_{\text{ODI}}$, step size $\eta_{\text{ODI}}$, number of restarts $N_R$
2: **Output:** Starting points $\{x_i^{start}\}$ for adversarial attacks
3: **for** $i = 1$ **to** $N_R$ **do**
4:     Sample $\mathbf{x}_0$ from $B(\mathbf{x}_{org})$, and sample $\mathbf{w}_{\text{d}} \sim U(-1, 1)^C$
5:     **for** $k = 0$ **to** $N_{\text{ODI}} - 1$ **do**
6:         $\mathbf{x}_{k+1} \leftarrow \text{Proj}_{B(\mathbf{x}_{org})}(x_k + \eta_{\text{ODI}} \, \text{sign}(\mathbf{v}_{\text{ODS}}(\mathbf{x}_k, \mathbf{f}, \mathbf{w}_{\text{d}})))$
7:     $\mathbf{x}_i^{start} \leftarrow x_{N_{\text{ODI}}}$

---

We also describe the algorithm of Boundary-ODS, used in Section 5.2 of the main paper. Algorithm B shows pseudo-code of Boundary-ODS. The original Boundary Attack [17] first sampled a random noise vector $\mathbf{q}$ from a Gaussian distribution $\mathcal{N}(\mathbf{0}, \mathbf{I})$ and then orthogonalized the vector to keep the distance from the original image (line 7 in Algorithm B). After that, the attack refined the vector $\mathbf{q}$ to reduce the distance from the original image such that the following equation holds:

$$d(\mathbf{x}, \mathbf{x}_{adv}) - d(\mathbf{x}, \mathbf{x}_{adv} + \mathbf{q}) = \epsilon \cdot d(\mathbf{x}, \mathbf{x}_{adv}) \tag{A}$$

where $d(a, b)$ is the distance between $a$ and $b$. We replace the random Gaussian sampling to ODS as in the line 5 and 6 of Algorithm B. Sampled vectors by ODS yield large changes for outputs on the target model and increase the probability that the updated image is adversarial (i.e. the image satisfies the line 9 of Algorithm B), so ODS makes the attack efficient.

---
**Algorithm B** Boundary Attack [17] with sampling update direction by ODS
---
1: **Input:** A targeted image $\mathbf{x}$, a label $y$, a target classifier $\mathbf{f}$, a set of surrogate models $\mathcal{G}$
2: **Output:** attack result $\mathbf{x}_{adv}$
3: Set the starting point $\mathbf{x}_{adv} = \mathbf{x}$ which is adversary
4: **while** $k <$ number of steps **do**
5:     Choose a surrogate model $\mathbf{g}$ from $\mathcal{G}$, sample $\mathbf{w}_{\text{d}} \sim U(-1, 1)^C$
6:     Set $\mathbf{q} = \mathbf{v}_{\text{ODS}}(\mathbf{x}_{adv}, \mathbf{g}, \mathbf{w}_{\text{d}})$
7:     Project $\mathbf{q}$ onto a sphere around the original image $\mathbf{x}$
8:     Update $\mathbf{q}$ with a small movement toward the original image $\mathbf{x}$ such that Equation (A) holds
9:     **if** $\mathbf{x}_{adv} + \mathbf{q}$ is adversarial **then**
10:        Set $\mathbf{x}_{adv} = \mathbf{x}_{adv} + \mathbf{q}$

---

# B Details of experiment settings

## B.1 Hyperparameters and settings for attacks in Section 4.1

We describe hyperparameters and settings for PGD and C&W attacks in Section 4.1.

Multiple loss functions $L(\cdot)$ can be used for PGD attacks, including the cross-entropy loss, and the margin loss defined as $\max_{i \neq y} f_i(\mathbf{x}) - f_y(\mathbf{x})$. We use the margin loss for PGD attacks to make considered attacking methods stronger.

PGD attacks have three hyperparameters: pertubation size $\epsilon$, step size $\eta$ and number of steps $N$. We chose $\epsilon = 0.3, 8/255, 4/255$, $\eta = 0.02, 2/255, 0.5/255$ and $N = 40, 20, 50$ for MadryLab (MNIST), MadryLab (CIFAR-10), ResNet152 Denoise (ImageNet), respectively. We use the whole test set except for ImageNet, where the first 1000 test images are used.

For C&W attacks, we define naïve random initialization to make sure the starting points are within an $\ell_2$ $\epsilon$-radius ball: we first sample Gaussian noise $\mathbf{w} \sim \mathcal{N}(\mathbf{0}, \mathbf{I})$ and then add the clipped noise

$\epsilon \cdot \mathbf{w}/\|\mathbf{w}\|_2$ to an original image. We set the perturbation radius of initialization $\epsilon$ by reference to attack bounds in other studies: $\epsilon = 2.0, 1.0, 5.0$ for MadryLab (MNIST), MadryLab (CIFAR-10), ResNet152 Denoise (ImageNet), respectively. we also set hyperparameters of C&W attacks as follows: max iterations are 1000 (MNIST) and 100 (CIFAR-10 and ImageNet), search step is 10, learning rate is 0.1, and initial constant is 0.01. The attack is performed for the first 1000 images (MNIST and CIFAR-10) and the first 500 images (ImageNet).

## B.2 Hyperparameter tuning for tuned ODI-PGD in Section 4.2

We describe hyperparameter tuning for our tuned ODI-PGD in Section 4.2. We summarize the setting in Table A.

Table A: Hyperparameter setting for tuned ODI-PGD in Section 4.2.

| model | ODI | | PGD | | |
| | total step $N_{\text{ODI}}$ | step size $\eta_{\text{ODI}}$ | optimizer | total step $N$ | step size (learning rate) $\eta_k$ |
| --- | --- | --- | --- | --- | --- |
| MNIST | 50 | 0.05 | Adam | 950 | 0.1    $(k < 475)$<br>0.01   $(475 \le k < 712)$<br>0.001 $(712 \le k)$ |
| CIFAR-10 | 10 | 8/255 | sign function | 140 | 8/255    $(k < 46)$<br>0.8/255   $(46 \le k < 92)$<br>0.08/255 $(92 \le k)$ |

For ODI, we increase the number of ODI step $N_{\text{ODI}}$ to obtain more diversified inputs than ODI with $N_{\text{ODI}} = 2$. In addition, we make step size $\eta_{\text{ODI}}$ smaller than $\epsilon$ on MNIST, because $\epsilon$-ball with $\epsilon = 0.3$ is large and $\eta_{\text{ODI}} = 0.3$ is not suitable for seeking the diversity within the large $\epsilon$-ball. In summary, we set $(N_{\text{ODI}}, \eta_{\text{ODI}}) = (50, 0.05), (10, 8/255)$ for the MNIST model and the CIFAR-10 model, respectively.

We tune hyperparameters of PGD based on Gowal et al. [16]. While several studies used the sign function to update images for the PGD attack, some studies [44, 16] reported that updates by Adam optimizer [45] brought better results than the sign function. Following the previous studies [44, 16], we consider the sign function as an optimizer and the choice of an optimizer as a hyperparameter. We use Adam for the PGD attack on the MNIST model and the sign function on the CIFAR-10 model.

We adopt scheduled step size instead of fixed one. Because we empirically found that starting from large step size brings better results, we set the initial step size $\eta_0$ as $\eta_0 = \epsilon$ on CIFAR-10. We update step size at $k = 0.5N, 0.75N$ on MNIST and $k = N/3, 2N/3$ for on CIFAR-10. When we use Adam, step size is considered as learning rate. Finally, we set PGD step $N$ as $N_{\text{ODI}} + N = 1000$ on MNIST and 150 on CIFAR-10.

## B.3 Setting for training on ImageNet in Section 5.3

We describe the setting of training of surrogate models on ImageNet in the experiment of Section 5.3. We use the implementation of training provided in PyTorch with default hyperparameters. Namely, training epochs are 90 and learning rates are changed depending on epoch: 0.1 until 30 epochs, 0.01 until 60 epochs, 0.001 until 90 epochs. Batch size is 256 and weight decay 0.0001.

## C  Additional results and experiments for ODI with white-box attacks

### C.1  Diversity offered by ODI

We empirically demonstrate that ODI can find a more diverse set of starting points than random uniform initialization, as pictorially shown in the left figures of Figure 1 of the main paper.

As an example of target models, we train a robust classification model using adversarial training [2] on CIFAR-10. We adopted popular hyperparameters for adversarial training under the $\ell_\infty$ PGD attack

on CIFAR-10: perturbation size $\epsilon = 8/255$, step size $\eta = 2/255$, and number of steps $N = 10$. Training epochs are 100 and learning rates are changed depending on epoch: 0.1 until 75 epochs, 0.01 until 90 epochs, and 0.001 until 100 epochs. Batch size is 128 and weight decay 0.0002.

On the target model, we quantitatively evaluate the diversity of starting points by each initialization in terms of pairwise distances of output values $\mathbf{f(x)}$. Each initialization is bounded within $\ell_\infty$ $\epsilon$-ball with $\epsilon = 8/255$. We pick 100 images on CIFAR-10 and run each initialization 10 times to calculate the mean pairwise distances among outputs for different starting points. As a result, the mean pairwise distance obtained from ODI is 6.41, which is about 15 times larger than that from uniform initialization (0.38). This corroborates our intuition that starting points obtained by ODI are more diverse than uniform initialization. We note that PGD does not generate diverse samples. When we use PGD with 2 steps as an initialization, the mean pairwise distance is only 0.43.

We also visualize the diversity offered by ODI. First, we focus on loss histogram of starting points by ODI and naïve uniform initialization. We pick an image from the CIFAR-10 test dataset and run each initialization 100 times. Then, we calculate loss values for starting points to visualize their diversity in the output space. The left panel of Figure A is the histogram of loss values for each initialization. We can easily observe that images from naïve initialization concentrate in terms of loss values (around $-1.0$), whereas images from ODI-2 (ODI with 2 steps) are much more diverse in terms of the loss values. We also observe that images from PGD-2 also take similar loss values. By starting attacks from these initial inputs, we obtain the histogram of loss values in the center panel of Figure A. We can observe that ODI-PGD generates more diverse results than PGD with naïve initialization (PGD-20).

In addition, we apply t-SNE [46] to the output logits for starting points by each initialization. We visualize the embedding produced by t-SNE in the right panel of Figure A. As expected, starting points produced by ODI are more diversified than those by naïve initialization.

Figure A: (Left): Histogram of loss values evaluated at starting points by ODI, naïve uniform initialization and PGD. PGD-2 means 2-step PGD with naïve initialization. The loss function is the margin loss. (Right) Histogram of loss values after attacks with 20 total steps. ODI-PGD-18 means 18-step PGD with 2-step ODI. (Right): Embedding for starting points sampled on each initialization produced by t-SNE.

## C.2 Analysis of the sensitivity to hyperparameters of ODI

For ODI, we mainly set the number of ODI steps $N_{\mathrm{ODI}} = 2$ and step size $\eta_{\mathrm{ODI}} = \epsilon$. To validate the setting, we confirm that ODI-PGD is not sensitive to these hyperparameters. We attack adversarially trained models on CIFAR-10 introduced in Section C.1, and adopt the same attack setup for ODI-PGD on CIFAR-10 as Section 4.1. We test $N_{\mathrm{ODI}} = 2, 4, 8, 16$ and $\eta_{\mathrm{ODI}} = \epsilon, \epsilon/2, \epsilon/4, \epsilon/8$, but exclude patterns with $N_{\mathrm{ODI}} \cdot \eta_{\mathrm{ODI}} < 2\epsilon$ to make $N_{\mathrm{ODI}} \cdot \eta_{\mathrm{ODI}}$ larger than or equal to the diameter of the $\epsilon$-ball. We calculate the mean accuracy for five repetitions of the attack, each with 20 restarts.

Table B shows the mean accuracy under ODI-PGD for different hyperparameters. The maximum difference in the mean accuracy among different hyperparameters of ODI is only 0.05%. Although large $N_{\mathrm{ODI}}$ and $\eta_{\mathrm{ODI}}$ will be useful to find more diversified starting points, the performance of ODI is not very sensitive to hyperparameters. Thus, we restrict $N_{\mathrm{ODI}}$ to a small value to give fair comparison in terms of computation time as much as possible. Table B also shows that the difference between the maximum and minimum accuracy is about 0.1% for all hyperparameter pairs. This result supports the stability of ODI.

Table B: The sensitivity to the number of ODI steps $N_{\text{ODI}}$ and step size $\eta_{\text{ODI}}$. We repeat each experiment 5 times to calculate statistics.

| $N_{\text{ODI}}$ | $\eta_{\text{ODI}}$ | mean | max | min |
|---|---|---|---|---|
| 2 | $\epsilon$ | 44.46% | 44.50% | 44.45% |
| 4 | $\epsilon/2$ | 44.47% | 44.50% | 44.42% |
| 4 | $\epsilon$ | 44.42% | 44.48% | 44.40% |
| 8 | $\epsilon/4$ | 44.47% | 44.52% | 44.44% |
| 8 | $\epsilon/2$ | 44.42% | 44.48% | 44.36% |
| 8 | $\epsilon$ | 44.46% | 44.49% | 44.42% |
| 16 | $\epsilon/8$ | 44.46% | 44.50% | 44.43% |
| 16 | $\epsilon/4$ | 44.46% | 44.50% | 44.40% |
| 16 | $\epsilon/2$ | 44.45% | 44.48% | 44.43% |
| 16 | $\epsilon$ | 44.44% | 44.47% | 44.41% |

## C.3 Accuracy curve for adversarial attacks with ODI

In Section 4, we experimentally represented that the diversity offered by ODI improved white-box $\ell_\infty$ and $\ell_2$ attacks. we describe the accuracy curve with the number of restarts for attacks with ODI and naïve initialization.

Figure B shows how the attack performance improves as the number of restarts increases in the experiment of Section 4.1. Attacks with ODI outperforms those with naïve initialization with the increase of restarts in all settings. These curves further corroborate that restarts facilitate the running of attack algorithms, and ODI restarts are more effective than naïve ones. We note that the first restart of ODI is sometimes worse than naïve initialization. It is because diversity can cause local optima, i.e. random directions of ODI are not always useful. With the increase of restarts, at least one direction is useful and the accuracy drops.

Figure B: The attack performance against number of restarts for attacks with ODI. (Top): the model accuracy for PGD, (Bottom): the average of minimum $\ell_2$ perturbations for C&W.

Next, we describe the accuracy curve for the comparison between state-of-the-are attacks and ODI-PGD in Section 4.2. To emphasize the stability of the improvement, we evaluate the confidence intervals of our results against MadryLab's MNIST and CIFAR-10 models. We run tuned ODI-PGD attack with 3000 restarts on MNIST and 100 restarts on CIFAR-10. Then, we sample 1000 runs on MNIST and 20 runs on CIFAR-10 from the results to evaluate the model accuracy, and re-sample 100 times to calculate statistics. Figure C shows the accuracy curve under tuned ODI-PGD. We observe that confidence intervals become tighter as the number of restarts grows, and tuned ODI-PGD consistently outperforms the state-of-the-art attack after 1000 restarts on MNIST and 20 restarts on CIFAR-10.

Figure C: Model accuracy under tuned ODI-PGD and the current state-of-the-art attacks [16]. The solid lines represent values from Table 2 and the error bars show 95% confidence intervals.

### C.4 Tighter estimation of robustness for various models

One important application of powerful adversarial attacks is to evaluate and compare different defense methods. In many previous works on defending against adversarial examples, PGD attack with naïve uniform initialization (called naïve-PGD) is a prevailing benchmark and its attack success rate is commonly regarded as a tight estimation on (worst-case) model robustness. In this section, we conduct a case study on six recently published defense methods [47, 48, 49, 50, 51, 52] to show that ODI-PGD outperforms naïve-PGD in terms of upper bounding the worst model accuracy under all possible attacks.

**Setup**    We use pre-trained models from four of those studies, and train the other two models [51, 52] using the settings and architectures described in their original papers. We run attacks with $\epsilon = 8/255$ on all test images. Other attack settings are the same as the experiment for CIFAR-10 in Section 4.1. Apart from comparing ODI-PGD and naïve-PGD, we also evaluate PGD attack without restarts (denoted as $PGD_1$) as it is adopted in several existing studies [47, 48, 49, 52].

Table C: Accuracy of models after performing ODI-PGD and naïve-PGD attacks against recently proposed defense models.

| model | (1) $PGD_1$ | (2) naïve-PGD | (3) ODI-PGD | (1)−(2) | (2)−(3) |
|---|---|---|---|---|---|
| UAT [47] | 62.63% | 61.93% | **57.43%** | 0.70% | 4.50% |
| RST [48] | 61.17% | 60.77% | **59.93%** | 0.40% | 0.84% |
| Feature-scatter [49] | 59.69% | 56.49% | **39.52%** | 3.20% | 16.97% |
| Metric learning [50] | 50.57% | 49.91% | **47.64%** | 0.56% | 2.27% |
| Free [51] | 47.19% | 46.39% | **44.20%** | 0.80% | 2.19% |
| YOPO [52] | 47.70% | 47.07% | **45.09%** | 0.63% | 1.98% |

**Results**    As shown in Table C, ODI-PGD uniformly outperforms naïve-PGD against all six recently-proposed defense methods, lowering the estimated model accuracy by 1–17%. In other words, ODI-PGD provides uniformly tighter upper bounds on the worst case model accuracy than naïve-PGD. Additionally, The accuracy ranking of the defence methods for ODI-PGD is different from naïve-PGD and $PGD_1$. These results indicate that ODI-PGD might be a better benchmark for comparing and evaluating different defense methods, rather than naïve-PGD and $PGD_1$.

## D    Additional results and experiments for ODS with black-box attacks

### D.1    Diversified samples by ODS

We empirically show that ODS can yield diversified changes in the output space of the target model, as shown in the right figures of Figure 1 of the main paper. Specifically, we evaluate the mean

pairwise distance among outputs for different perturbations by ODS and compare it with the distance among outputs for random Gaussian sampling.

We use pre-trained Resnet50 [28] and VGG19 [29] model as the target and surrogate models, respectively. We pick 100 images on ImageNet validation set and sample perturbations 10 times by each sampling method. For comparison, we normalize the perturbation to the same size in the input space. Then, the obtained pairwise distance on the target model by ODS is 0.79, which is 10 times larger than the pairwise distance by random Gaussian sampling (0.07). This indicates that the diversity by ODS is transferable.

## D.2    Success rate curve in Section 5.1 and Section 5.2

In Section 5.1, we demonstrated that SimBA-ODS outperformed state-of-the-art attacks in terms of the query-efficiency. As an additional result, we give the success rate curve of score-based attacks with respect to the number of queries in the experiments. Figure D shows how the success rate changes with the number of queries for SimBA-ODS and SimBA-DCT for the experiment of Table 3. SimBA-ODS especially brings query-efficiency at small query levels. In Figure E, we also describe the success rate curve for the experiment of Table 5. ODS-RGF outperforms other methods in the $\ell_2$ norm.

untargeted                                    targeted

Figure D: Relationship between success rate and number of queries for score-based SimBA-ODS and SimBA-DCT.

untargeted ($\ell_2$)          targeted ($\ell_2$)          untargeted ($\ell_\infty$)          targeted ($\ell_\infty$)

Figure E: Relationship between success rate and number of queries for SimBA-ODS, ODS-RGF, and Square Attack. Each attack is evaluated with norm bound $\epsilon = 5(\ell_2), 0.05(\ell_\infty)$.

In Section 5.2, we demonstrated that Boundary-ODS outperformed state-of-the-art attacks in terms of median $\ell_2$ perturbation. Here, we depict the relationship between the success rate and perturbation size (i.e. the frequency distribution of the perturbations) to show the consistency of the improvement. Figure F describes the cumulative frequency distribution of $\ell_2$ perturbations for each attack at 10000 queries. Boundary-ODS consistently decreases $\ell_2$ perturbations compared to other attacks in both untargeted and targeted settings.

## D.3    Comparison of ODS with TREMBA

We run experiments to compare ODS with TREMBA, which is a state-of-the-art attack with surrogate models, as we mentioned in Section 5.1.2. TREMBA leverages surrogate models to learn a low-dimensional embedding so as to obtain initial adversarial examples using a transfer-based attack and then update them using a score-based attack. Although TREMBA uses random sampling, ODS does not work well with TREMBA because random sampling of TREMBA is performed in the embedding space. In addition, it is difficult to directly compare attacks with ODS (e.g., ODS-RGF)

Figure F: Cumulative frequency distribution of $\ell_2$ perturbations at 10000 queries for decision-based attacks.

and TREMBA because we do not discuss the combination of ODS with transfer-based attacks in this paper.

However, we can start attacks with ODS (e.g., ODS-RGF) from images generated by any transfer-based attacks and compare the attack with TREMBA. We generate starting points by SI-NI-DIM [35] (Scale-Invariant Nesterov Iterative FGSM integrated with diverse input method), which is a state-of-the-art transfer-based attack, and run ODS-RGF from these starting points.

We adopt the same experiment setup as TREMBA [34]: we evaluate attacks against four target models (VGG19, ResNet34, DenseNet121, MobileNetV2) for 1000 images on ImageNet and use four surrogate models (VGG16, Resnet18, Squeezenet [53] and Googlenet [54] ). We set the same hyperparameters as in the original paper [34] for TREMBA. For fair comparisons, we set the same sample size (20) and use the same surrogate models as TREMBA for ODS-RGF. We also set step size of ODS-RGF as 0.001. As for SI-NI-DIM, we set hyperparameters referring to the paper [35]: maximum iterations as 20, decay factor as 1, and number of scale copies as 5. We describe results in Table D. We can observe that ODS-RGF with SI-NI-DIM is comparable to TREMBA.

We note that ODS is more flexible than TREMBA in some aspects. First, TREMBA is specific in the $\ell_\infty$-norm, whereas ODS can be combined with attacks at least in $\ell_\infty$ and $\ell_2$-norms. In addition, TREMBA needs to train a generator per target class in targeted settings, whereas ODS does not need additional training.

Table D: Comparison of ODS-RGF with TREMBA against four target models. The first two rows and bottom two rows describe results for untargeted (U) attacks and targeted (T) attacks, respectively. Targeted class for targeted attacks is class 0.

|  | attack | VGG19 | | ResNet34 | | DenseNet121 | | MobilenetV2 | |
| --- | --- | --- | --- | --- | --- | --- | --- | --- | --- |
|  |  | success | query | success | query | success | query | success | query |
| U | TREMBA [34] | **100.0%** | 34 | **100.0%** | 161 | **100.0%** | 157 | **100.0%** | 63 |
|  | SI-NI-DIM [35] + ODS-RGF | **100.0%** | **18** | 99.9% | **47** | 99.9% | **50** | **100.0%** | **29** |
| T | TREMBA [34] | 98.6% | 975 | 96.7% | **1421** | **98.5%** | **1151** | **99.0%** | **1163** |
|  | SI-NI-DIM [35] + ODS-RGF | **99.4%** | **634** | **98.7%** | 1578 | 98.2% | 1550 | 98.3% | 2006 |

## D.4 Performance of ODS against different target models

In this paper, we used pre-trained ResNet50 model as the target model for all experiments in Section 5. Here we set pre-trained VGG19 model as the target model and run experiments to show that the efficiency of ODS is independent with target models. As surrogate models, we replace VGG19 with ResNet50, i.e. we use four pre-trained models (ResNet50, ResNet34, DenseNet121, MobileNetV2).

We run experiments for SimBA-ODS in Section 5.1 and Boundary-ODS in Section 5.2. All settings except the target model and surrogate models are the same as the previous experiments. In Table E and F, ODS significantly improves attacks against VGG19 model for both SimBA and Boundary Attack. This indicates that the efficiency of ODS does not depend on target models.

Table E: Query counts and $\ell_2$ perturbations for score-based Simple Black-box Attacks (SimBA) against pre-trained VGG19 model on ImageNet.

| | num. of surrogates | untargeted | | | targeted | | |
|---|---|---|---|---|---|---|---|
| attack | | success rate | average query | median $\ell_2$ distance | success rate | average query | median $\ell_2$ distance |
| SimBA-DCT [18] | 0 | **100.0%** | 619 | 2.85 | **100.0%** | 4091 | 6.81 |
| SimBA-ODS | 4 | **100.0%** | **176** | **1.35** | 99.7% | **1779** | **3.31** |

Table F: Median $\ell_2$ perturbations for decision-based Boundary Attacks against pre-trained VGG19 model on ImageNet.

| | | number of queries | | | | | |
|---|---|---|---|---|---|---|---|
| | num. of surrogates | untargeted | | | targeted | | |
| attack | | 1000 | 5000 | 10000 | 1000 | 5000 | 10000 |
| Boundary[17] | 0 | 45.62 | 11.79 | 4.19 | 75.10 | 41.63 | 27.34 |
| Boundary-ODS | 4 | **6.03** | **0.69** | **0.43** | **24.11** | **5.44** | **2.97** |

## D.5 Effect of the choice of surrogate models

In Section 5.1 and 5.2, we mainly used four pre-trained models as surrogate models. To investigate the effect of the choice of surrogate models, we run attacks with seven different sets of surrogate models. All settings except surrogate models are the same as the previous experiments.

Table G and H shows results for SimBA-ODS and Boundary-ODS, respectively. First, the first four rows in both tables are results for a single surrogate model. The degree of improvements depends on the model. ResNet34 gives the largest improvement and VGG19 gives the smallest improvement. Next, the fifth and sixth rows show results for sets of two surrogate models. By combining surrogate models, the query efficiency improves, especially for targeted attacks. This means that the diversity from multiple surrogate models is basically useful to make attacks strong. Finally, the performances in the seventh row are results for four surrogate models, which are not always better than results for the combination of two models (ResNet34 and DenseNet121). When the performances for each surrogate model are widely different, the combination of those surrogate models could be harmful.

Table G: Query counts and $\ell_2$ perturbations for SimBA-ODS attacks with various sets of surrogate models. In the column of surrogate models, R:ResNet34, D:DenseNet121, V:VGG19, M:MobileNetV2.

| surrogate models | num. | untargeted | | | targeted | | |
|---|---|---|---|---|---|---|---|
| | | success rate | average query | median $\ell_2$ distance | success rate | average query | median $\ell_2$ distance |
| R | 1 | **100.0%** | 274 | 1.35 | 95.3% | 5115 | 3.50 |
| D | 1 | **100.0%** | 342 | 1.38 | 96.7% | 5282 | 3.51 |
| V | 1 | **100.0%** | 660 | 1.78 | 88.0% | 9769 | 4.80 |
| M | 1 | **100.0%** | 475 | 1.70 | 95.3% | 6539 | 4.53 |
| R,D | 2 | **100.0%** | **223** | **1.31** | 98.0% | **3381** | **3.39** |
| V,M | 2 | **100.0%** | 374 | 1.60 | 96.3% | 4696 | 4.27 |
| R,V,D,M | 4 | **100.0%** | 241 | 1.40 | **98.3%** | 3502 | 3.55 |

In Section 5.1.2, we compared ODS-RGF with P-RGF only using the ResNet34 surrogate model. To show the effectiveness of ODS-RGF is robust to the choice of surrogate models, we evaluate ODS-RGF with different surrogate models. Table I shows the query-efficiency of ODS-RGF and P-RGF with the VGG19 surrogate model. We can observe that ODS-RGF outperforms P-RGF for all settings and the results are consistent with the experiment in Section 5.1.2.

Table H: Median $\ell_2$ perturbations for Boundary-ODS attacks with various sets of surrogate models. In the column of surrogate models, R:ResNet34, D:DenseNet121, V:VGG19, M:MobileNetV2.

| surrogate models | num. | number of queries | | | | | |
| | | untargeted | | | targeted | | |
| | | 1000 | 5000 | 10000 | 1000 | 5000 | 10000 |
|---|---|---|---|---|---|---|---|
| R | 1 | 9.90 | 1.41 | 0.79 | 31.32 | 11.49 | 7.89 |
| D | 1 | 10.12 | 1.39 | 0.76 | 32.63 | 11.30 | 7.44 |
| V | 1 | 22.68 | 3.47 | 1.52 | 49.18 | 24.26 | 17.75 |
| M | 1 | 20.67 | 2.34 | 1.10 | 44.90 | 18.62 | 12.01 |
| R,D | 2 | **7.53** | 1.07 | 0.61 | **26.00** | 8.08 | 6.22 |
| V,M | 2 | 17.60 | 1.70 | 0.92 | 39.63 | 14.97 | 9.21 |
| R,V,D,M | 4 | 7.57 | **0.98** | **0.57** | 27.24 | **6.84** | **3.76** |

Table I: Comparison between ODS-RGF and P-RGF with the VGG19 surrogate model. Settings in the comparison are the same as Figure 4.

| norm | attack | num. of surrogates | untargeted | | | targeted | | |
| | | | success | average queries | median $\ell_2$ perturbation | success | average queries | median $\ell_2$ perturbation |
|---|---|---|---|---|---|---|---|---|
| $\ell_2$ | RGF | 0 | **100.0%** | 633 | 3.07 | **99.3%** | 3141 | 8.23 |
| | P-RGF [25] | 1 | **100.0%** | 467 | 3.03 | 97.0% | 3130 | 8.18 |
| | ODS-RGF | 1 | **100.0%** | **294** | **2.24** | 98.0% | **2274** | **6.60** |
| $\ell_\infty$ | RGF | 0 | 97.0% | 520 | - | 25.0% | 2971 | - |
| | P-RGF [25] | 1 | 98.7% | 337 | - | 29.0% | 2990 | - |
| | ODS-RGF | 1 | **99.7%** | **256** | - | **45.7%** | **2116** | - |

## D.6 Effect of the number of surrogate models for the experiment in Section 5.3

We described that surrogate models with limited out-of-distribution training dataset are still useful for ODS in Section 5.3. In the experiment, we used five surrogate models with the same ResNet18 architecture. Here, we reveal the importance of the number of surrogate models through experiments with the different number of models. Table J shows the result for Boundary-ODS with the different number of surrogate models. With the increase of the number of models, the query efficiency consistently improves.

Table J: Median $\ell_2$ perturbations for Boundary-ODS attacks with different number of surrogate models against out-of-distribution images on ImageNet.

| num. of surrogates | number of queries | | | | | |
| | untargeted | | | targeted | | |
| | 1000 | 5000 | 10000 | 1000 | 5000 | 10000 |
|---|---|---|---|---|---|---|
| 1 | 19.45 | 2.90 | 1.66 | 47.86 | 25.30 | 20.46 |
| 2 | 15.45 | 2.42 | 1.35 | 43.45 | 19.30 | 13.78 |
| 3 | 13.75 | 1.96 | 1.14 | **41.63** | 16.91 | 11.14 |
| 4 | 14.23 | 1.86 | 1.21 | 41.65 | 14.86 | 9.64 |
| 5 | **11.27** | **1.63** | **0.98** | 41.67 | **13.72** | **8.39** |

## D.7 Score-based attacks with ODS against out-of-distribution images

In Section 5.3, we demonstrated that the decision-based Boundary-ODS attack works well even if we only have surrogate models trained with limited out-of-distribution dataset. Here, we evaluate score-based SimBA-ODS with these surrogate models. Except surrogate models, we adopt the same setting as Section 5.1.

In Table K, SimBA-ODS with out-of-distribution dataset outperforms SimBA-DCT in untargeted settings. In targeted settings, while SimBA-ODS improves the $\ell_2$ perturbation, the average queries for SimBA-ODS are comparable with SimBA-DCT. We hypothesize that it is because ODS only explores the subspace of the input space. The restriction to the subspace may lead to bad local optima. We can mitigate this local optima problem by applying random sampling temporally when SimBA-ODS fails to update a target image in many steps in a low.

We note that decision-based Boundary-ODS with OOD dataset is effective, as shown in Section 5.3. We hypothesize that the difference in effectiveness is because Boundary-ODS does not use scores of the target model and thus does not trap in local optima.

Table K: Query counts and $\ell_2$ perturbations for SimBA-ODS attacks with surrogate models trained with OOD images on ImageNet.

| | untargeted | | | targeted | | |
|---|---|---|---|---|---|---|
| attack | success rate | average queries | median $\ell_2$ perturbation | success rate | average queries | median $\ell_2$ perturbation |
| SimBA-DCT [18] | **100.0%** | 909 | 2.95 | **97.0%** | 7114 | 7.00 |
| SimBA-ODS (OOD dataset) | **100.0%** | **491** | **1.94** | 94.7% | **6925** | **4.92** |
| SimBA-ODS (full dataset) | 100.0% | 242 | 1.40 | 98.3% | 3503 | 3.55 |

### D.8 ODS against robust defense models

In this paper, we mainly discuss transferability when surrogate models and the target model are trained with similar training schemes. On the other hand, it is known that transferability decreases when models are trained with different training schemes, e.g. the target model uses adversarial training and surrogate models use natural training. If all surrogate models are trained with natural training scheme, ODS will also not work against adversarially trained target models. However, we can mitigate the problem by simultaneously using surrogates obtained with various training schemes (which are mostly publicly available). In order to confirm this, we run an experiment to attack a robust target model using SimBA-ODS with both natural and robust surrogate models (a natural model and a robust model). In Table L, the first row shows the attack performance of SimBA-DCT (without surrogate models) and the others show the performance of SimBA-ODS. In the fourth row of Table L, SimBA-ODS with natural and robust surrogate models significantly outperforms SimBA-DCT without surrogate models. This suggests that if the set of surrogates includes one that is similar to the target, ODS still works (even when some other surrogates are "wrong"). While the performance with natural and robust surrogate models slightly underperforms single adversarial surrogate model in the third row, dynamic selection of surrogate models during the attack will improve the performance, as we mentioned in the conclusion of the paper.

Table L: Transferability of ODS when training schemes of surrogate models are different from the target model. R50 shows pretrained ResNet50 model, and R101(adv) and R152(adv) are adversarially trained ResNeXt101 and ResNet152 denoise models from [31], respectively. All attacks are held in the same setting as Section 5.1.

| target | surrogate | success rate | average queries | median $\ell_2$ perturbation |
|---|---|---|---|---|
| R101(adv) | - | 89.0% | 2824 | 6.38 |
| R101(adv) | R50 | 80.0% | 4337 | 10.15 |
| R101(adv) | R152(adv) | **98.0%** | **1066** | **4.93** |
| R101(adv) | R50, R152(adv) | **98.0%** | 1304 | 5.62 |

## E Relationship and Comparison between ODS and MultiTargeted

In this section, we describe that ODS gives better diversity than the MultiTargeted attack [16] for initialization and sampling.

MultiTargeted is a variant of white-box PGD attacks, which maximizes $f_t(\mathbf{x}) - f_y(\mathbf{x})$ where $\mathbf{f}(\mathbf{x})$ is logits, $y$ is the original label and $t$ is a target label. The target label is changed per restarts. In other words, MultiTargeted moves a target image to a particular direction in the output space, which is represented as like $\mathbf{w}_d = (1, 0, -1, 0)$ where 1 and -1 correspond to the target and original label, respectively. Namely, the procedure of MultiTargeted is technically similar to ODS.

However, there are some key differences between MultiTargeted and ODS. One of the difference is the motivation. MultiTargeted was proposed as a white-box attack and the study only focused on $\ell_p$-bounded white-box attacks. On the other hand, our study gives broader application for white- and black-box attacks. As far as we know, ODS is the first method which exploits the output diversity for initialization and sampling.

Another difference is the necessity of the original label of target images. ODS does not require the original class of the target image, and thus ODS is applicable for black-box attacks even if surrogate models are trained with out-of-distribution training dataset, as shown in Section 5.3. On the other hand, since MultiTargeted exploits the original label of target images to calculate the direction of the attack, we cannot apply MultiTargeted to sampling for black-box attacks against out-of-distribution images.

Finally, the level of diversity is also different. As we mentioned in Section 6, the direction of MultiTargeted is restricted to away from the original class. This restriction could be harmful for diversity because the subspace to explore directions is limited. To show this statement, we apply MultiTargeted to initialization for white-box attacks and sampling for black-box attacks, and demonstrate that ODI provides better diversity than MultiTargeted for initialization and sampling (especially for sampling).

**Initialization in white-box settings**   We apply MultiTargeted to initalization for white-box attacks in Section 4.1. Table M represents the comparison of the attack performance with initialization by MultiTargeted and ODI. For PGD attacks, MultiTargeted is slightly better than ODI. We hypotheses that it is because MultiTargeted was developed as a variant of PGD attacks and the initialization by MultiTargeted also works as an attack method. On the other hand, ODI outperforms MultiTargeted for C&W attacks. In this setting, MultiTargeted does not work as an attack method, and thus the difference in the diversity makes the difference in the performance.

Table M: Comparison of model performance under attacks with MultiTargeted (MT) and ODI. The values are model accuracy (lower is better) for PGD and the average of the minimum $\ell_2$ perturbations (lower is better) for C&W. All results are the average of three trials. Results for ODI are from Table 1.

| | PGD | | C&W | |
| model | MT | ODI | MT | ODI |
|---|---|---|---|---|
| MNIST | $\mathbf{89.95} \pm 0.05\%$ | $90.21 \pm 0.05\%$ | $2.26 \pm 0.01$ | $\mathbf{2.25} \pm 0.01$ |
| CIFAR-10 | $\mathbf{44.33} \pm 0.01\%$ | $44.45 \pm 0.02\%$ | $0.69 \pm 0.01$ | $\mathbf{0.67} \pm 0.00$ |
| ImageNet | $\mathbf{42.2} \pm 0.0\%$ | $42.3 \pm 0.0\%$ | $2.30 \pm 0.01$ | $\mathbf{1.32} \pm 0.01$ |

**Sampling in black-box settings**   We use MultiTargeted for sampling on the Boundary Attack in Section 5.2 (called Boundary-MT), and compare it with Boundary-ODS. Table N and Figure G show the results of the comparison. While Boundary-MT outperforms the original Boundary Attack, Boundary-ODS finds much smaller adversarial perturbation than Boundary-MT.

In Figure G, Boundary-MT slightly outperforms Boundary-ODS at small queries. We hypotheses that it is because MultiTargeted not works for providing diversity, but works for the approximation of gradients of the loss function. However, with the number of queries, the curve of Boundary-MT is saturated, and Boundary-MT underperforms Boundary-ODS. This is an evidence that the restriction of directions is harmful for sampling.

Table N: Median $\ell_2$ perturbations for Boundary Attack with ODS and MultiTargeted (MT).

| attack | number of queries | | | | | |
|---|---|---|---|---|---|---|
| | untargeted | | | targeted | | |
| | 1000 | 5000 | 10000 | 1000 | 5000 | 10000 |
| Boundary [17] | 45.07 | 11.46 | 4.30 | 73.94 | 41.88 | 27.05 |
| Boundary-ODS | **7.57** | **0.98** | **0.57** | **27.24** | **6.84** | **3.76** |
| Boundary-MT | 7.65 | 2.20 | 2.01 | 28.16 | 18.48 | 16.59 |

Untargeted                    Targeted

Figure G: Relationship between median $\ell_2$ perturbations and the number of queries for Boundary Attack with ODS and MultiTargeted. Error bars show 25%ile and 75%ile of $\ell_2$ perturbations.