[Reviews · NeurIPS 2020]

Review 1

Summary and Contributions: This paper proposes a novel sampling strategy (ODS) to improve the efficacy of randomness in white- and black-box attacks (i.e., to initialize gradient-based white-box attacks or generate update directions in black-box attacks). ODS provides maximal diversity among generated samples by randomly sampling based on output distance and improves white-box by quality by a very small amount (with some caveats that I describe in weaknesses). The results on black-box queries are more compelling, e.g., reducing query cost for ImageNet attacks by 2x, which is significant.

Strengths: Applies to both white-box and black-box. Well motivated (see Figure 1). Tested on white-box attacks, surrogate models, score attacks, decision attacks Whitebox results show minor improvement, but blackbox results (especially Table 5) are quite compelling.

Weaknesses: The authors mention that ODS cost is roughly equivalent to doing one attack iteration of PGD (lines 101-102). It seems in that case, a fairer comparison would be to compare a standard PGD-(k+1) attack with a PGD-k attack under ODS. In that case, the differences in Table 1 and Table 2 may disappear. It is not clear why ODS for, say, PGD-10 should outperform PGD-11, since the latter is optimizing for maximum loss in any direction where ODS is optimizing for loss in a specific direction. I would like Table 1 and Table 2 include such a comparison. Requires a surrogate model for decision and score-based attacks (although section 5.3 alleviates this by showing success with OOD images) Improves a part of existing attacks rather than introducing a new one, so a bit less novelty / impact I did not get the complexity comparison. What is 42 x 20, for instance, under ODI-PGD in Table 2. How about showing execution time instead or in addition? How does that compare?

Correctness: Seems correct, but the comparison in Table 1 and Table 2 could include a comparable PGD attack that takes the same time to generate. But I could not judge the claims on complexity (e.g., 50 times lower complexity on line 172) since it is not clear what complexity measures and how it relates to execution time for different attack strategies.

Clarity: Generally well-written, flow from motivation to existing processes to the changes made is good. The flow between sections 4.1 and 4.2 could be improved, however, as the term ODI-PGD is introduced in 4.2, making it sound like a new thing, when it is just the attack mentioned by 4.1. The complexity numbers in Table 2 could also be explained better. It is hard to tell where the numbers are coming from and the caption says sum while the table lists products.

Relation to Prior Work: The authors acknowledge adversarial training, generative models, regularization techniques, certified defenses. The authors also do a good job of framing where their changes to existing techniques. Related work touches upon surrogate models and MultiTargeted attacks, explaining how their technique is different.

Reproducibility: Yes

Additional Feedback: The authors make a nice observation that random perturbations, which are critical for random restarts and gradient direction sampling, often use naive uniform or Gaussian distributions in the input pixel space. Such techniques may not lead to diverse samples in the output space, where we want diversity of samples. The net effect seems, for white-box attacks, to add an iterative optimization process before the original attack. The authors find that only 2 steps are required for enough diversity, which is good for efficiency purposes. The black-box attack does require a surrogate model, as it measures the diversity of gradient directions on a surrogate, but is still interesting nonetheless. The authors provide some evidence that surrogate ODS diversity transfers to the target model in Figure 1, albeit to a lesser extent. That seems to be expected, however. The most interesting result, which seems to be noted by the authors, is in improved black-box efficiency. Improvements in white-box attacks seem modest, but in line with the small part of the process being changed. Section 5.3 was a good inclusion. A natural question to the applicability of ODS was how to get the surrogate model, and showing that it still gained some improvement while having only OOD images greatly answers this limitation. As stated in the conclusion, using unlabeled datasets would be a nice next step. Typo: ODI-OGD on line 159


Review 2

Summary and Contributions: The paper proposes a method, ODS, to sample points x around a given input x_orig such that, for a classifier f, the vectors f(x) and f(x_orig) are sufficiently different according to some metric. Such technique is used in two scenarios: for white-box attacks, points yielding diversified output can be used as starting points and are more effective than randomly sampled points, while for black-box attacks ODS is used to sample update directions, improving success rate and query efficiency of such attacks.

Strengths: - The method is simple and can be applied to improve many existing attacks. - The authors show improvements due to the proposed method through extensive experiments, on white-box, score-based and decision-bases attacks. - The method could make easier and more precise the evaluation of robustness.

Weaknesses: - In the white-box scenario, the improvements provided by ODI are one side consistent, but on the other side often of small entity. Moreover, according to Sec. C.3 it requires multiple restarts (at least 2) to get better results than the standard initialization. - In the black-box scenario, ODS exploits multiple surrogate models, unlike the original methods to which it is compared. Then it is expected that such extra knowledge improves the performance of the attacks. A better comparison would be with other attacks using a surrogate model.

Correctness: The method seems correct, and empirical evaluation clear and quite comprehensive.

Clarity: The paper is well written and presents clearly the results.

Relation to Prior Work: The authors make an effort to compare the proposed methods to prior works for what concerns white-box attacks, while dismiss the experimental comparison to other black-box attacks.

Reproducibility: Yes

Additional Feedback: - To test the diversity of the points generated by ODI (L106), the authors compare it to uniformly random sampling. However, it is known that random noise usually doesn't change the classification of a point (see e.g. Szegedy et al. [1]), and then it seems likely that the output of such perturbed points is still similar to the original one. Conversely, the points given by ODI are the results of a 2 steps optimization scheme. Then, how is the diversity of the points provided by 2 steps of e.g. standard PGD on the cross-entropy loss, or CW loss? This comparison would, in my opinion, help to better understand the effectiveness of the proposed method. - As mentioned above, the usage of (many) surrogate models for the black-box attacks makes the direct comparison to Square Attack, Boundary Attack, HopSkipJump and Sign-OPT a bit problematic, as the threat model is different (different level of knowledge). ODS definitely improves the performance of SimBA and Boundary Attack, but I think a better comparison would be with attacks using also a surrogate models, like [23, 25]. - Also the authors say that such methods [23, 25] might suffer if OOD images are used to train the surrogate model. While I agree with such statement, I guess that the same would hold true for ODS if the training scheme is different, e.g. the target model uses adversarial training, the surrogate models natural training. Do the authors have an intuition about this? Overall, I think the idea of diversifying the starting points to improve standard attacks is valid, especially since the proposed technique is simple and applicable to many existing attacks (this could also be validated with further experiments). For the black-box attacks, I think the further comparison mentioned above would significantly strengthen the paper. I'd be happy to raise my score in case the previous points are properly addressed. ### Update post rebuttal ### I'm satisfied with the replies about surrogate models with different training schemes and the output diversity given by PGD-2 (I suggest to add also the pairwise distance as reported in the paper). The comparison with black-box attacks using surrogate models, P-RGF in this case, is pretty convincing and I think shows that ODS can improve also other methods. I still think that also [23] should be included in such comparison: while it's true that [23] focus on Linf, untargeted attacks can be performed with a single generator (not one per target class) which is publicly available. As mentioned by R3, I'd also suggest to explicitly discuss which types of black-box attacks ODS can be integrated with. Then I raise the original score (to 7, accept), since I think the contribution is meaningful and the experiments extensive and that the paper can still be improved by adding the further experiments presented in the rebuttal.


Review 3

Summary and Contributions: The paper focuses on improving white-box and black-box adversarial attacks using the so-called “output diversification” approach. The main idea is based on the observation that the simple and widely used random initialization of the starting point in adversarial attacks does not lead to diverse outputs of the classifier in the logits space. Instead, the authors suggest to use essentially a random objective in the logit space in order to find a good initialization for a subsequent adversarial attack in the white-box setting, or as a search direction for black-box attacks based on random sampling of new directions. This general idea of “output diversification” leads to improvements both in the white-box setting (marginally better than the MultiTargeted attack) and in the black-box setting (significantly outperforming the state of the art).

Strengths: - The idea works for both white-box and black-box attacks which shows that the phenomenon is quite general. The improvement is particularly significant for black-box attacks (particularly, decision-based). - The output diversity claim is empirically justified. - The experiment in “Sec. 5.3 Effectiveness of ODS with out-of-distribution images” is nice to justify the overall knowledge model of the attacker used in this paper, i.e. that surrogate models are available. Particularly because many recent black-box attacks that rely on surrogate models just directly use models trained on full ImageNet, often even with very similar hyperparameters. - It is good to see detailed ablation studies about the choice of the surrogate models in the appendix.

Weaknesses: - There are also other black-box attacks that also rely on surrogate models. In particular, [a] is the most prominent example from the recent literature. [a] have code available online, thus such a comparison would be interesting to see for the rebuttal. - What happens if as a search direction w_d one samples a vector aligned with the true class (e.g., a one-hot vector where the non-zero element corresponds to the correct class)? My understanding is that this will lead to maximization of the confidence in the correct class. Would that be also helpful as an output diversification approach? I can imagine that if the answer is no, this can be harmful for the case when there are only a few classes. Would such an initialization essentially mean a waste of computations / queries? - The improvement over the MultiTargeted attack in the white-box setting is only marginal (although, seems to be consistent). - It would be good to provide a clear discussion why you decided to combine SimBA and Boundary attack with ODS, but not the Square Attack and HopSkipJump attacks which outperform SimBA and Boundary attacks. Currently, this is not clear from the paper. - It is unclear to me whether it is possible to extend the ODS black-box attack approach to other Lp-norms, for example to Linf. Currently, it seems that the idea of sampling an ODS direction is quite specific to the attacks based on sampling a random direction such as SimBA and the Boundary Attack. A further discussion about this would be beneficial to provide. [a] Improving Black-box Adversarial Attacks with a Transfer-based Prior, NeurIPS 2019, https://arxiv.org/abs/1906.06919

Correctness: Yes.

Clarity: Yes.

Relation to Prior Work: Yes. In particular, the relation to the MultiTargeted attack is covered in detail which is beneficial since there are similarities between MultiTargeted and this work.

Reproducibility: Yes

Additional Feedback: Lines 153-154: “When the non-linearity of a target model is strong, the difference in diversity between the input and output space could be large” -- I think that words like “strong non-linearity” should be avoided as they are too hand-wavy and it is not even clear what they mean. I think it’s okay to use them but only if a measure of nonlinearity is clearly defined. Lines 196-197: “Namely, the diversity afforded by ODS is also helpful to find natural adversarial examples.” -- Unclear to me what is meant by “natural adversarial examples”. Line 256: “restricted to away” -- There is a missed word here. Line 261-262: “In contrast, ODS does not approximate gradients of the loss function, but instead focuses on the diversity of outputs for the target model.” -- This statement is a bit unclear to me since ODS is a meta approach and it does not perform gradient approximation as long as the base method (e.g., SimBA) does not do it. Line 289: I think there should be a comma instead of a full stop. I think it would be beneficial to mention clearly in the experimental part that all the methods you currently compare to (SimBA-DCT, Square Attack, Boundary attack, HopSkipJump attack, Sign-OPT) *do not* assume an extra knowledge such as having surrogate models. Just to be clear that in those experiments, your method has a clear advantage which comes from a more permissive knowledge model of the attacker. Table F: the horizontal line in the table should extend until the end. Title: "can be" in the title should be capitalized as these are also verbs. ------- Update after the rebuttal ------------ In general, I'm satisfied with the rebuttal and agree to raise the score from 6 to 7. But I would urge the authors to improve the experimental evaluation of the black-box setting in the final version (in case of acceptance). In particular, with respect to the following points: 1.) A comparison to [25] as done in the rebuttal with a few additions. I'm happy to see that the proposed method can be also useful for Linf attacks. But there I would recommend to: - Benchmark the transfer to ResNet-50 from a less similar model than ResNet-34. E.g. by taking VGG or Inception as a surrogate model. - Include the results of the Linf Square Attack that has similar query efficiency to the proposed method (73 avg. queries vs 74; see Table 2 in https://arxiv.org/pdf/1912.00049.pdf) while not relying on any surrogate models. 2.) A comparison to [23] as suggested by R2. 3.) I agree with R2 that: "the usage of (many) surrogate models for the black-box attacks makes the direct comparison to Square Attack, Boundary Attack, HopSkipJump and Sign-OPT a bit problematic, as the threat model is different (different level of knowledge)" and would suggest to explicitly have a column in Tables 3 and 4 (extended by the experiments from the rebuttal) that would clearly show how many surrogate models each method uses. Just to make it clear to the reader that one is comparing methods with **different knowledge models** with respect to surrogate models.

[Author Response · NeurIPS 2020]

We thank all reviewers for their insightful feedback. We are encouraged they find ODS to be simple (R2), well-motivated
(R1), applicable to many existing attacks (R2) including both white- and black-box attacks (R1,3), and evaluated with
extensive experiments (R2) which show significant improvements in black-box attacks (R1,3) and justifications of
surrogate models (R3). We address some specific comments below and will incorporate all feedback received.

@R2,3 – "Comparison with black-box attacks using surrogate models would strengthen the paper." Great suggestion!
We focus on [25] which R2 and R3 cited. [25] proposed P-RGF which uses prior knowledge to estimate the gradient of
the target model more efficiently than RGF. RGF uses random sampling to estimate the gradient, so we can combine
ODS with RGF and compare it with P-RGF under $\ell_2$ and $\ell_\infty$ norms (results in Table i below) . **The average number**
**of queries required by ODS-RGF is smaller than P-RGF ([25]) in all settings**. It suggests ODS-RGF can estimate
the gradient more precisely than P-RGF by exploiting diversity obtained from surrogate models. R2 also cited [23].
While we did not have enough time for an additional experimental comparison, we note that [23] is specific to the $\ell_\infty$
norm and needs to train a generator per target class, which is quite restricted compared to ODS.

Table i: Comparison of ODS-RGF and P-RGF for 300 images on ImageNet. The target and surrogate models are pre-trained ResNet50 and ResNet34 models, respectively. As for hyperparameters, the number of max queries is 10000, sample size is 10, step size is 0.5 ($\ell_2$) and 0.005 ($\ell_\infty$), and epsilon is $\sqrt{0.001 \cdot 224^2 \cdot 3}$ ($\ell_2$) and 0.05 ($\ell_\infty$).

| norm | attack | untargeted | | | targeted | | |
|---|---|---|---|---|---|---|---|
| | | success | queries | $\ell_2$ perturbation | success | queries | $\ell_2$ perturbation |
| $\ell_2$ | RGF | **100.0%** | 633 | 3.07 | **99.3%** | 3141 | 8.23 |
| | P-RGF [25] | **100.0%** | 211 | 2.08 | 97.0% | 2296 | 7.03 |
| | ODS-RGF | **100.0%** | **133** | **1.50** | **99.3%** | **1043** | **4.47** |
| $\ell_\infty$ | RGF | 97.0% | 520 | - | 25.0% | 2971 | - |
| | P-RGF [25] | 99.7% | 88 | - | 65.3% | 2123 | - |
| | ODS-RGF | **100.0%** | **74** | - | **92.0%** | **985** | - |

@R2 – "Does ODS suffer from differences in the training scheme (e.g. adversarially and naturally)?" Yes, partly. That
being said, we can mitigate the problem by simultaneously using surrogates obtained with various training schemes
(which are mostly publicly available). We run a new experiment to attack a robust target model using SimBA-ODS
with both natural and robust surrogate models (a natural model and a robust model). SimBA-ODS still outperforms
SimBA-DCT without surrogate models (e.g. average query is 1304 vs 2824). This suggests that if the set of surrogates
includes one that is similar to the target, ODS still works (even when some other surrogates are "wrong").

@R3 – "What happens if a search direction of ODS is a vector aligned with the true class?" It might accelerate attacks
but make the perturbation large due to less diversity (A related phenomenon is shown in Figure G for MultiTargeted).

@R3 – "Why do you combine SimBA and Boundary attack with ODS?" A reason is these attacks use random sampling.
Another one is popularity. These attacks are common benchmarks.

@R1 – "Comparison under the same step for Table 1 and 2 would be fairer." We agree and performed new experiments
with ODI-PGD-(k-2), which outperforms PGD-k in Table 1 on all datasets (90.21% vs 90.31% on MNIST, 44.45% vs
46.06% on CIFAR-10 and 42.3% vs 43.5% on ImageNet). For Table 2, we also ran tuned ODI-PGD with 1000 total
steps on MNIST and can confirm the result in a single run is within the confidence intervals in Figure C.

@R2 – "Comparison of diversity between 2 steps of ODI and PGD would be helpful." We compare diversity like in
Figure A. In the left panel of Figure i, losses for points generated by ODI-2 (2 steps of ODI) are more diverse than
PGD-2. This diversity also brings diversity in attack results after 20 steps (see the right panel).

Figure i: Histogram of loss values after some update steps. Each attack runs 100 times for one sample image. PGD-2 and PGD-20 are initialized by naïve uniform initialization. The loss function is the margin loss.

@R1 – "What is computational complexity in Table 2?" It is the number of gradient computations, e.g. 42 steps $\times$ 20
restarts = 840. We confirmed the wall-clock time for calculation of a ODI step is the same as PGD.

[Meta-Review · NeurIPS 2020]

This paper proposes an efficient and decently general method for improving upon white and black box adversarial attacks, showing good empirical results. The reviews were initially a bit mixed, but thanks to an effective rebuttal and a satisfactory discussion between reviewers, a consensus emerged according to which the paper is undoubtedly of an acceptable standard for NeurIPS. I invite the authors to take special heed of the comments made by the reviewers (especially post-rebuttal comments by R3) in preparing the final version of this paper, as it seems like there is room for improvement based on the feedback given.